# HOTAIR Participation in Glycolysis and Glutaminolysis Through Lactate and Glutamate Production in Colorectal Cancer

**DOI:** 10.3390/cells14050388

**Published:** 2025-03-06

**Authors:** Laura Cecilia Flores-García, Verónica García-Castillo, Eduardo Pérez-Toledo, Samuel Trujano-Camacho, Oliver Millán-Catalán, Eloy Andrés Pérez-Yepez, Jossimar Coronel-Hernández, Mauricio Rodríguez-Dorantes, Nadia Jacobo-Herrera, Carlos Pérez-Plasencia

**Affiliations:** 1Unidad de Biomedicina, Facultad de Estudios Superiores Iztacala, Universidad Nacional Autónoma de México (UNAM), Tlalnepantla 54090, Mexico; lcfloresg@hotmail.com (L.C.F.-G.); garciaver@gmail.com (V.G.-C.); eduardo.perez.toledo02@gmail.com (E.P.-T.); 2Laboratorio de Genómica, Instituto Nacional de Cancerología, Av. San Fernando 22, Belisario Domínguez Secc 16, Tlalpan, Mexico City 14080, Mexico; samuel.trujano1@gmail.com (S.T.-C.); oliver.millan.sg@gmail.com (O.M.-C.); eperezy2306@gmail.com (E.A.P.-Y.); jossithunders@gmail.com (J.C.-H.); 3Experimental Biology PhD Program, DCBS, Universidad Autónoma Metropolitana-Iztapalapa, Mexico City 09340, Mexico; 4Laboratorio de Oncogenómica, Instituto Nacional de Medicina Genómica, Mexico City 14610, Mexico; mrodriguez@inmegen.gob.mx; 5Unidad de Bioquímica, Instituto Nacional de Ciencias Medicas y Nutrición Salvador Zubirán (INCMNSZ), Mexico City 14080, Mexico; nadia.jacobo@gmail.com

**Keywords:** metabolism, cancer, HOTAIR, glycolysis, glutaminolysis and colorectal cancer

## Abstract

Metabolic reprogramming plays a crucial role in cancer biology and the mechanisms underlying its regulation represent a promising study area. In this regard, the discovery of non-coding RNAs opened a new regulatory landscape, which is in the early stages of investigation. Using a differential expression model of HOTAIR, we evaluated the expression level of metabolic enzymes, as well as the metabolites produced by glycolysis and glutaminolysis. Our results demonstrated the regulatory effect of HOTAIR on the expression of glycolysis and glutaminolysis enzymes in colorectal cancer cells. Specifically, through the overexpression and inhibition of HOTAIR, we determined its influence on the expression of the enzymes PFKFB4, PGK1, LDHA, SLC1A5, GLUD1, and GOT1, which had a direct impact on lactate and glutamate production. These findings indicate that HOTAIR plays a significant role in producing “oncometabolites” essential to maintaining the bioenergetics and biomass necessary for tumor cell survival by regulating glycolysis and glutaminolysis.

## 1. Introduction

According to data from GLOBOCAN (2022) [1], colorectal cancer is a public health problem worldwide as it ranks third among tumors with the highest incidence, with 1,926,425 new cases, and second in mortality, with 904,019 cancer-related deaths. This disease is characterized by the uncontrolled proliferation of the epithelium of the large intestine or rectum, leading to the formation of a tumor that can affect adjacent tissues or distal organs, such as the liver, lungs, or peritoneum [1,2,3].

On the other hand, in Hanahan and Weinberg’s description of the hallmarks of cancer, metabolic reprogramming was identified as one of the fundamental characteristics of the tumor cell. The role of cellular metabolic reprogramming focuses on increasing or suppressing the activity of metabolic pathways to produce “oncometabolites” that support cell survival and proliferation [4,5]. Furthermore, this metabolic reprogramming focuses on three processes: altered bioenergetics, enhanced biosynthesis, and redox balance [4].

Among metabolic pathways that undergo reprogramming in tumorigenesis, glycolysis and glutaminolysis play a significant role in generating intermediates to meet energy demand and the synthesis of macromolecules [4]. The overactivation of signaling pathways is associated with the transcriptional and transductional modulation of metabolism-related proteins, such as pyruvate kinase 2 (PKM2), glucose transporter 1 (GLUT1 or SLC2A1), lactate dehydrogenase A (LDHA), hexokinase 1 (HK1), and glutamate dehydrogenase (GLUD), among others [4,6].

In this sense, the PI3K/Akt/HIF-1α pathway is one of the most important signaling pathways in regulating tumor metabolism through the activity of HIF-1α as a transcription factor [4,5]. In turn, it has been identified that HIF-1α functions as an interconnecting molecule between metabolic reprogramming and epigenetic regulation processes since it has been determined that some long non-coding RNAs (lncRNAs) can play a role in metabolic reprogramming through the regulation of HIF-1α, such as lncRNA-p21, HISLA, and MALAT [6,7].

The lncRNAs are RNAs with a size greater than 200 nucleotides, which have complex secondary and tertiary structures that result in a wide variety of regulatory mechanisms when interacting with proteins, DNA, or RNA; exerting functions as decoys, scaffolds, and competitive endogenous RNAs, among others; and acting as epigenetic regulators [6,8]. An example of this type of molecule is the lncRNA Homeobox Transcript Antisense Intergenic RNA (HOTAIR), which has been described as regulating the expression of SLC2A1 and glutaminase (GLS) [9,10].

The lncRNA HOTAIR has a size of 2.2 kb and is transcribed from the antisense strand of the *HOXC* genes located on the long arm of Chromosome 12, which is a transacting polyadenylated intergenic lncRNA [11,12,13]. HOTAIR is composed of seven exons and functions primarily as a molecular decoy, sequestering microRNAs (miRNAs) and RNA-binding proteins, regulating chromatin state and, thus, gene transcriptional silencing through its binding to polycomb repressor complex 2 (PRC2) and lysine-specific demethylase 1 (LSD1), favoring the trimethylation of histone H3 at lysine 27 (H3K27me3) [12,14,15,16]. However, its role in the transcriptional regulation of other molecules involved in both glucose and glutamine metabolism is unknown.

In this study, we used a differential expression model of HOTAIR in colorectal cancer cells to evaluate the role of this lncRNA in metabolic reprogramming and its potential interaction with HIF-1α. Our findings indicate that HOTAIR regulates the expression of various enzymes involved in glycolytic and glutaminolytic metabolism in tumor cells. A specific mechanism by which it functions involves its interaction with HIF-1α, which is linked to the induction of LDHA and GLUD1. This interaction has a direct effect on the production of lactate and glutamate. Our findings demonstrate that HOTAIR plays a fundamental role in the reprogramming of two key metabolic pathways in tumor cells, highlighting its potential as a therapeutic target against cancer.

## 2. Materials and Methods

### 2.1. Cell Lines

The commercial colorectal cell lines CRL-1459, HCT-116, RKO, and SW-620 were obtained from the Functional Genomics Laboratory of the National Cancer Institute of México. HCT-116 cells were cultured in McCoy’s 5A (Merck, medium supplemented with 10% fetal bovine serum. SW-620 and RKO cells were cultured in DMEM/F-12 medium supplemented with 10% fetal bovine serum. Immortalized non-tumor colonic epithelial cells, CRL-1459, were cultured in EMEM medium with 10% fetal bovine serum. Incubation was carried out at 37 °C in a 5% CO_2_ atmosphere.

### 2.2. HOTAIR Knockdown and Overexpression

To inhibit HOTAIR expression, HOTAIR DsiRNAs and TriFECTa^®^ kits (Integrated DNA Technologies, Inc. (IDT), IA, USA) were used in a combination of three DsiRNAs at individual concentrations of 10 nM. For HOTAIR overexpression, we used a pcDNA 3.1 HOTAIR construct from GeneArt Thermo (Thermo-Fisher, Waltham, MA, USA) at 2.5 μg. Transfections were performed using Lipofectamine 3000 (Invitrogen, Waltham, MA, USA) according to the manufacturer’s instructions. RNA extraction was performed 48 h post-transfection (Invitrogen, Thermo-Fisher, Waltham, MA, USA).

### 2.3. Proliferation Assay 

After 24 h of transfection, cells were seeded at a density of 22,500 cells/cm^2^. The following day, the medium was changed to one containing 1% FBS and left for another 24 h. For the triple therapy assay (3Tx), the same number of cells were seeded and, after 24 h, the medium was changed to 1% FBS. The medium was then removed and a drug combination was added at the following concentrations: 1.44 µM of Doxorubicin, 43 mM of Metformin, and 36.64 mM of Sodium Oxamate, allowing it to act for 24 h. For both assays, after the 24-h treatment period, the medium was removed and the cells were washed with PBS. Then, 100 µL of medium containing MTT (0.5 mg/mL) was added to each well and incubated for 4 h at 37 °C. Following this incubation, the medium was removed and 100 µL of 0.5% DMSO was added for 20 min at room temperature. The optical density was measured at 570 nm using an Epoch microplate spectrophotometer (Bio Tek, Agilent Technologies, Santa Clara, CA, USA).

### 2.4. RNA Extraction, Reverse Transcription, and qRT-PCR

RNA extraction was performed using TRIzol™ reagent (Invitrogen, Thermo-Fisher, Waltham, MA, USA), according to the manufacturer’s instructions. Reverse mRNA transcription was performed using the High-Capacity cDNA Reverse Transcription kit (Applied Biosystems, Thermo-Fisher, Waltham, MA, USA). Additionally, qRT-PCR was performed using the PowerUp™ SYBR™ Green Master Mix kit (Applied Biosystems, Thermo-Fisher, Waltham, MA, USA), with SFRS9 or GAPDH as the normalizing constituent. The relative expression of all targets evaluated was calculated using the 2^−ΔΔCt^ method. 

The sequences of the primers used are presented in the following Table 1.

### 2.5. RNA Binding Protein Immunoprecipitation (RIP) Assay

Interaction probability between HOTAIR and HIF-1α was analyzed using the RPISeq software version 1 (http://pridb.gdcb.iastate.edu/RPISeq/, accessed 10 May 2024). RNA binding protein immunoprecipitation (RIP) with a specific antibody to HIF-1α (Invitrogen, Thermo-Fisher, Waltham, MA, USA, PA116601) was conducted using the Magna RIP™ RNA-binding protein immunoprecipitation kit (17–704, EMD Millipore, Burlington, MA, USA). For this assay, SW-620, HCT-116, and RKO cell lines were lysed by RIP lysis buffer, after which the lysate was incubated on magnetic beads associated with protein G, conjugated with 5 μg to the antibody and negative control normal rabbit IgG (Merck Millipore, Burlington, MA, USA). Samples were incubated with proteinase K and RNA was isolated. Finally, to corroborate the interaction of HOTAIR with HIF-1 α, qRT-PCR was performed.

### 2.6. Measurements of Glucose, Lactate, Glutamine/Glutamate, and ATP Levels

In untreated cells, metabolism assays were performed by culturing 10,000 cells per well in a 96-well plate. After 24 h, the cells were changed to MEM medium with 1% FBS and left for 4 h. For cells with HOTAIR overexpression or silencing, after 24 h of transfection, 5000 cells per well were cultured and allowed to incubate for 24 h, after which the medium was changed and allowed to incubate for 4 h. Extracellular glucose concentration was determined by the Glucose-Glo™ Assay kit (Promega Corporation, Cat. J6021, Madison, WI, USA) as described by the manufacturer. Extracellular lactate levels were quantified using the Lactate-Glo™ Assay kit (Promega Corporation, Cat. J5021, WI, USA) as described by the manufacturer. Intracellular ATP concentration was evaluated by the CellTiter-Glo™ Assay kit (Promega Corporation, Cat. G7570, Madison, WI, USA) as described by the manufacturer. The intracellular Glutamine/Glutamate level was determined using the Glutamine/Glutamate-Glo™ kit (Promega Corporation, Cat. J8021, WI, USA) as described by the manufacturer.

### 2.7. Tissue Expression for In Silico Meta-Analysis

Expression data of HOTAIR, HIF1A, SLC2A1, HK1, HK2, HK3, GPI, PFKP, TPI1, GAPDH, PGK1, PGK2, PGAM1, PGAM2, PKLR, LDHA, LDHB, LDHC, SLC1A5, SLC38A3, GLS2, GLUD1, and GOT1 were obtained from The Cancer Genome Atlas in different stages of 471 colorectal cancer samples vs. 41 non-tumor samples. The protein expression of HIF1A, SLC2A1, HK1, HK2, HK3, GPI, PFKP, TPI1, GAPDH, PGK1, PGK2, PGAM1, PGAM2, PKLR, LDHA, LDHB, LDHC, SLC1A5, SLC38A3, GLS2, GLUD1, and GOT1 was assessed using antibody-based proteomic data from the Human Protein Atlas.

### 2.8. Visualization of the Protein–Protein Interaction Network 

We used the online analysis tool STRING (https://string-db.org/, accessed 19 August 2024 with a confidence of 0.150, to construct the PPI network diagram for the proteins of our interest. Cytoscape software Version 3.10.3 was used to build the interaction network map.

### 2.9. HIF-1α and HOTAIR Co-Localization Assay

To detect the co-localization of HIF-1α and HOTAIR, we performed an in situ hybridization assay. In this process, 20,000 cells were seeded in 12-well plates. The cells were fixed with 4% paraformaldehyde for 30 min and subsequently hybridized with the HOTAIR Stellaris^®^ FISH probe (Human HOTAIR CAL Fluor RED 590 Dye, VSMF-2176–5, LGC Bioserarch Technologies, Hoddesdon, UK) at a dilution of 1:100 for 6 h at 37 °C. Following this, the cells were incubated overnight at 4 °C with an antibody against HIF-1α (Invitrogen, Thermo-Fisher, Waltham, MA, USA PA116601) at a dilution of 1:100. Afterward, the cells were treated with a secondary antibody against rabbit Alexa Fluor^®^ 488 (Abcam, ab150077, Cambridge, UK). Finally, the nuclei were stained with Hoechst at a dilution of 1:3000 (YB2820692). Images of the prepared samples were captured using the LSM 7 DUO confocal microscope (ZEISS, Oberkochen, Germany) provided by the National Cancer Institute. Image preparation and placement testing were performed using FIJI software Version 2.9.0 and the JaCop plugin v2.0.

### 2.10. Chromatin Immunoprecipitation (ChIP) Assay 

To validate that HOTAIR regulates the expression of metabolic enzymes through HIF-1a, we performed a chromatin immunoprecipitation assay (Merck Millipore, MA, USA) following the manufacturer’s protocol. The differential expression model previously described in RKO, HCT-116, and SW-620 cells was used. Protein–DNA cross-linking was performed with formaldehyde (1%) and sonication for 30 s and four pulses. Immunoprecipitation of the complexes was performed with the anti-HIF-1α antibody (Invitrogen, Thermo-Fisher, Waltham, MA, USA PA116601) specific for ChIP assays. The immunoprecipitated DNA sequences for *SLC2A1*, *LDHA*, *SLC1A5*, and *GLUD1* were amplified by PCR under standard conditions and then analyzed by electrophoresis on a 1.5% agarose gel.

### 2.11. Database Search 

We examined overall survival rates to investigate the relationship between HOTAIR or *HIF1A* expression levels and survival status in colorectal tumors, employing the Kaplan–Meier analysis in the GEPIA2 database (http://gepia2.cancer-pku.cn, accessed 24 July 2024). Additionally, we conducted a co-expression analysis of HOTAIR and *HIF1A* in colorectal cancer tissue using the Starbase Encori tool (https://starbase.sysu.edu.cn/index.php, accessed 24 July 2024).

### 2.12. Statistical Analysis 

All values are expressed as mean ± SEM. Data were analyzed in the IBM SPSS Statistics 26 software using one-way ANOVA followed by LSD or the Tamhane multiple comparison test, as appropriate.

## 3. Results

### 3.1. Evaluation of HOTAIR and HIF-1α Expression in Colorectal Cancer

To evaluate the role of HOTAIR in metabolic reprogramming, the first step was to determine its expression level in colorectal tissue. In this sense, we observed that HOTAIR is overexpressed in tumor tissue (9.31-fold, *p* ≤ 0.005) compared to healthy tissue (Figure 1A) and, in turn, increases according to the clinical stage (Figure 1B). Meanwhile, Kaplan–Meier analysis showed that HOTAIR overexpression was a poor prognosis for patient survival (Figure 1C). On the other hand, in colorectal cancer lines, qPCR analysis corroborated the increase in HOTAIR expression in tumor cells (RKO, HCT-116, and SW-620) compared to non-tumor cells (CRL-1459) (Figure 1D).

In addition to the above, as previously described, HIF-1α plays an important role as a mediator of the regulatory effect of lncRNAs on tumor metabolism [17,18]. In the case of *HIF1A*, a small decrease in its expression levels is observed in tumor tissue vs. normal tissue (0.07 times, *p* ≤ 0.05) (Figure 1E), in addition to its null effect on patient survival (Appendix A). Meanwhile, the protein level was higher in tumor tissue compared to normal tissue (Figure 1F) and, similarly, the expression level of HIF1A was higher in HCT-116 and SW-620 cells (3.78 and 17 times (*p* ≤ 0.05), respectively) compared to non-tumor cells (Figure 1G). Furthermore, a correlation is shown to exist between the expression levels of HOTAIR and HIF1A in colorectal cancer cells (correlation coefficient 0.904, *p* ≤ 0.005) (Figure 1H) but this measurement is not observed in patients (Appendix A).

The above data suggested an association between HOTAIR and HIF-1α. Therefore, the protein–RNA interaction analysis predicted a high probability of binding between both molecules (Figure 1I), which was corroborated by the RIP immunoprecipitation assay, which demonstrated that the SW-620 and RKO lines presented a significant enrichment in the association between HOTAIR and HIF1α (0.67-fold and 8.31-fold compared to IgG, respectively) (Figure 1J).

The results suggested that high levels of HOTAIR may be linked to the advancement of colorectal cancer in both patients and cell lines. Additionally, there seems to be a significant connection between HOTAIR and HIF-α, indicating the potential involvement of HOTAIR in cellular metabolism.

### 3.2. Probability of Regulation of Proteins Involved in Glycolysis and Glutaminolysis by the HIF1α/HOTAIR Axis

Metabolic adaptation is known to be largely coordinated by the transcription factor HIF-1α through the presence of Hypoxia Response Elements (HREs) in the promoters of glycolytic genes [4,19]. Furthermore, lncRNAs have been reported to bind to proteins and modify their function [20], suggesting that a possible mechanism for HOTAIR involvement in metabolic regulation is through its binding to HIF-1α to enhance its function as a transcription factor.

From protein–protein interaction modeling, we predicted that glycolysis and glutaminolysis enzymes might exhibit HIF-1α regulation with a high interaction score (LDHA, SLC2A1, phosphoglycerate kinase 1 (PGK1), glyceraldehyde-3-phosphate dehydrogenase (GAPDH), and HK2), as well as a medium interaction with phosphoglycerate mutase (PGAM1), 6-Phosphofructo-2-Kinase/Fructose-2,6-Bisphosphatase 4 (PFKFB4), glucose-6-phosphate isomerase (GPI), phosphoglycerate kinase (PGK), and triosephosphate isomerase (TPI1), whereas among glutaminolysis enzymes, only solute carrier family 1 member 5 (SLC1A5) showed a stronger interaction (Figure 2A,B). However, our predictions regarding the number of hypoxia response elements (HREs) indicated a high probability of a regulatory role of HIF-1α in all glycolytic and glutaminolytic enzymes tested (Figure 2B and Appendix A).

This regulatory process was confirmed when we determined the mRNA expression levels of these enzymes in colorectal cancer cells vs. non-tumor cells where HK2, PFKFB4, aldose c (ALDOC), pyruvate kinase (PKLR), LDHA, pyruvate dehydrogenase (PDHA), SLC1A5, GLS2, GLUD1, and aspartate aminotransferase 1 (GOT1) were overexpressed (Figure 2C,D). While in patients and immunohistochemical analyses, the enzymes that presented a higher level of expression in tumor tissue were SLC2A1, HK2, ALDOC, PFKP, TPI1, GAPDH, PGK1, LDHA, SLC1A5, GLS2, and GOT1 (Appendix A). By performing a correlation analysis, we determined that PGK1, PKLR, LDHA, SLC1A5, and GLUD1 showed a positive correlation with the expression level of HOTAIR (Figure 2E).

These findings indicated that both HIF-1α and HOTAIR could participate in the transcriptional regulation of metabolic enzymes, playing a crucial role in metabolic reprogramming.

### 3.3. HOTAIR Promotes Glycolysis and Glutaminolysis Enzyme Expression

HOTAIR has been reported to regulate the expression of the transcription factor *HIF1A*, which regulates the synthesis of SLC2A1 [6,21]. In addition to the above, our results supported the role of the HOTAIR/HIF-1α axis. Therefore, to corroborate this effect on the expression of glucose and glutamine metabolism enzymes, we generated a differential expression model of HOTAIR in colorectal cancer lines (Figure 3A).

When evaluating cell viability in the RKO line, we observed 50 and 75% increases by the overexpression and inhibition of HOTAIR, respectively, compared to Vector or Scrambled. For the HCT-116 line, cell viability decreased by 40 and 15% due to the increase and reduction in HOTAIR, respectively. In the SW-620 line, the overexpression of HOTAIR reduced viability by 50%, contrary to the 25% increase induced by the silencing of lncRNA (Figure 3B).

On the other hand, a differential effect on the expression of enzymes involved in glucose and glutamine metabolism was observed, depending on the cell line. In the RKO line, modifications in HOTAIR expression increased the mRNA levels of PFKFB4, GAPDH, LDHA, SLC1A5, GLUD1, and GOT1 (Figure 3C,D, Appendix A). In contrast, in the HCT-116 cell line, the effect of HOTAIR was limited to the regulation of PGK1 and LDHA expression (Figure 3C,D, Appendix A). While in the SW-620 line, HOTAIR modulation impacted the expression of SLC2A1, HK2, PFKB4, PGK1, PGAM1, LDHA, SLC1A5, GLUD1, and GOT1 (Figure 3C,D, Appendix A).

By observing the discrepancies between the different colorectal cancer cell lines, we considered that the enzymes that were affected by both HOTAIR overexpression and inhibition in at least two of the three lines evaluated would be considered HOTAIR targets. Thus, using a Venn diagram, we identified that the enzymes regulated by HOTAIR were PFKFB4, PGK1, LDHA, SLC1A5, GLUD1, and GOT1 (Figure 3E).

The data enabled us to identify the enzymes involved in glycolysis and glutaminolysis that were regulated by HOTAIR yet were not linked to an increase in HIF1A mRNA expression.

### 3.4. HOTAIR Regulates Lactate and Glutamate Production in Colorectal Cancer Cells

Current evidence has demonstrated the positive effect of HOTAIR in increasing glucose consumption and lactate production in hepatocarcinoma cells [22,23]. In addition, HOTAIR has so far only been confirmed to be involved in the expression of SLC2A1, HK2, and LDHA [22]. However, several of these enzymes possess isoforms that can substitute their function and that, in turn, can be regulated by HIF-1α, such as SLC2A1 and GLUT3 (SLC2A3), HK1 and HK2, and others [24]. Therefore, it was necessary to evaluate the impact of HOTAIR on the production of glycolysis and glutaminolysis metabolites.

To assess glucose consumption, we measured its concentration in the extracellular medium. Thus, an increase in glucose levels, compared to the vector/scrambled control, indicates a reduction in carbohydrate uptake. For the RKO line, there was an increase in glucose consumption, both in the overexpression and in the silencing of HOTAIR (Figure 4A, left panel). In the HCT-116 line, although the overexpression led to a decrease in the expression of SLC2A1, an increase in glucose consumption was observed (Figure 4A, middle panel). In contrast, the SW-620 cell line did not show significant changes in glucose consumption associated with the modulation of HOTAIR expression, despite the positive regulation of SLC2A1 (Appendix A).

Lactate production was increased after HOTAIR overexpression in all three lines used. However, HOTAIR silencing only affected the production of this metabolite in the RKO and SW-620 lines (Figure 4B, left and right panels). This observation is in agreement with our previous findings on the impact of HOTAIR on LDHA mRNA expression, both endogenously and after the establishment of the differential expression pattern (Figure 2C and Figure 3C).

On the other hand, anaerobic glycolysis is the primary pathway for ATP production in tumor cells. Given the regulation of lactate production induced by HOTAIR, we anticipated a similar effect on ATP production. However, the HCT-116 and SW-620 cell lines exhibited a decrease in ATP levels, both with HOTAIR overexpression and knockdown (Figure 4C, middle and right panels). In contrast, the RKO cell line demonstrated a decrease in ATP production by HOTAIR overexpression while silencing the lncRNA increased ATP levels (Figure 4C, left panel).

Although, in the differential expression model, we had identified the SLC1A5 transporter as one of the enzymes regulated by HOTAIR, the results obtained from glutamine consumption showed a decrease both by overexpression and knockdown of HOTAIR in the RKO and SW-620 lines (Figure 4D). While in the HCT-116 line, silencing of the lncRNA induced an increase in glutamine uptake (Figure 4D, central panel). Further, glutamate production was regulated proportionally to HOTAIR expression in the three colorectal cancer lines, where overexpression of the lncRNA increased glutamate production and silencing promoted the inverse effect (Figure 4E).

These results emphasize the role of HOTAIR in regulating glucose and glutamine metabolism by influencing glycolytic and glutaminolytic enzymes, such as LDHA and GLUD1, which are linked to the production of lactate and glutamate. It is important to note that the increase in glutamate production is also mediated by GLS but HOTAIR did not regulate GLS2 in either the HCT-116 or RKO cell lines (Appendix A). Therefore, the enzymatic transformation may involve an isoform of this enzyme that is, in turn, regulated by HOTAIR [25] since glutamate production was directly associated with lncRNA expression.

### 3.5. The Role of Hif-1α in the Transcriptional Regulation of Metabolic Enzymes Modulated by HOTAIR

Although our results from the differential expression model showed the ability of HOTAIR to induce the synthesis of metabolic enzymes, it was necessary to determine whether its mechanism of action was exerted through the physical interaction between HOTAIR and HIF-1α that had been previously observed (Figure 1J) since the expression of this transcription factor was only regulated in the SW-620 line (Figure 3C).

Using a co-localization assay, we observed that HOTAIR presents a differential distribution depending on the cell line. In the case of the RKO cell line, HOTAIR showed a mainly nuclear distribution, with only low points of co-localization with HIF-1α (Figure 5A upper panels). This finding was supported by the calculation of the overlap coefficient of regions of interest (ROIs), which calculated a value of 0.4, indicating that the presence of HOTAIR presents a 40% co-localization pattern with HIF-1α (Figure 5B). This is contrary to the perinuclear cytoplasmic distribution observed in the SW-620 cell line (Figure 5A lower panels), where there was a higher abundance of HOTAIR and the overlap coefficient with HIF-1α was 0.6 (Figure 5B). In the case of the HCT-116 line, the distribution of HOTAIR was nuclear, forming localization foci and low overlap with HIF-1α (Appendix A). 

Subsequently, using a ChIP assay targeting *LDHA*, *SLC1A5*, and *GLUD1*, in addition to *SLC2A1*, which has been described as a HIF-1α target [26], we assessed the positioning of HIF-1α at the promoters of HOTAIR-regulated metabolic enzymes. Our findings revealed that HIF-1α was enriched at the promoters of *LDHA* and *GLUD1* only in the RKO line, as well as there being an increase in its position after HOTAIR overexpression (Figure 5C, left panel). Meanwhile, the HOTAIR overexpression favored the positioning of HIF-1α at the promoters of *LDHA* in the SW-620 line and *SLC2A1* in the HCT-116 line, but not under endogenous conditions (Figure 5C, right panel, and Appendix A).

Together, these results indicate that, although HOTAIR and HIF-1α interact, this binding is only important for the regulation of *LDHA* and *GLUD1* in the RKO cell line endogenously, despite presenting a low expression of HOTAIR. This is contrary to the secondary role that this regulation in the SW-620 line, even though HOTAIR has an essential regulatory effect on metabolism that seems to be independent of HIF-1α.

### 3.6. Effect of the Pharmacological Inhibition of HOTAIR on Colorectal Cancer Cells

After confirmation that HOTAIR participates in the regulation of aberrant metabolism, it opens an opportunity window as a therapeutic target for the treatment of cancer. Therefore, based on previous studies of our work group, the use of a pharmacological combination based on Doxorubicin/Metformin/Sodium Oxamate (3Tx) was proposed, which has demonstrated antitumor effectiveness in several in vitro and in vivo models of breast and colorectal cancer through the inhibition of PI3K/Akt/mTOR/HIF-1α signaling [27,28,29].

Based on the regulation of HIF-1α and the large number of enzymes regulated by HOTAIR, we decided to evaluate 3Tx in the SW-620 cell line. Thus, when performing a viability assay with the pharmacological combination, we observed a 54% reduction while HOTAIR expression was reduced 0.5-fold compared to control or untreated cells (Ctr) (Figure 6A,B). Meanwhile, the expression of the previously identified enzymes that were regulated by HOTAIR also decreased after treatment with 3Tx and, in particular, the glycolysis enzymes were those that presented the greatest change in their expression (Figure 6C).

After confirming that 3Tx could influence the expression of HOTAIR and its glycolytic and glutaminolysis targets, we investigated the potential synergistic effect of HOTAIR knockdown in combination with 3Tx treatment. We observed that 3Tx treatment resulted in a 50% reduction in cell viability compared to control cells (Ctr) and a 57% reduction when combined with HOTAIR silencing (Figure 6D).

Concerning these findings, we hypothesized the possible involvement of other transcription factors associated with metabolic reprogramming, among which c-MYC stands out [4,6]. To explore this, we conducted a bioinformatics analysis and found that c-MYC is overexpressed at both the gene and protein levels in tumor tissue compared to non-tumor tissue, which impacts patient survival (Appendix A). Additionally, our prediction of the interaction between HOTAIR and c-MYC indicated a moderate binding probability. The protein–protein interaction model also suggested the potential regulation of several metabolic proteins, including GAPDH, LDHA, SLC2A1, HK2, PGK1, and SLC1A5 (Appendix A). These findings provide insights into the metabolic role of HOTAIR through an alternative mechanism involving c-MYC, warranting further investigation.

At the metabolic level, glucose consumption was lowered by exposure to 3Tx while, in conjunction with HOTAIR silencing, it induced an increase in carbohydrate uptake (Figure 6E). In contrast, while lactate production remained unchanged with 3Tx exposure, it did decrease when HOTAIR was silenced (Figure 6F). Regarding ATP production, exposure to 3Tx led to a reduction in ATP levels and this decrease became more significant when HOTAIR was silenced (Figure 6G). Conversely, the use of 3Tx increased glutamine consumption and glutamate production; however, this effect was reversed when HOTAIR knockdown occurred (Figure 6H,I).

These results indicate that HOTAIR is an important therapeutic target for cancer treatment as its silencing, combined with a pharmacological strategy, leads to metabolic changes that affect cell viability.

## 4. Discussion

Cancer is a complex disease primarily linked to genetic alterations, such as mutations, deletions, and amplifications. However, the discovery of non-coding RNAs has broadened our understanding of transcriptional and post-translational regulation as these molecules play essential roles in various cellular functions [30]. In particular, long non-coding RNAs (lncRNAs) have emerged as regulatory molecules in the tumor cell and, despite the identification of several transcribed RNAs of this class, their function within carcinogenesis still needs to be characterized in depth.

Part of the regulatory function of lncRNAs in the tumor cell has been associated with the regulation of oncogenes and tumor suppressor genes to induce proliferation and metastasis, as well as the identification of their role in aberrant metabolism [31,32]. This effect has been described mainly due to their capacity as molecular scaffolds, such as in the case of MALAT and lncRNA-p21 that regulate glycolysis [6,7]. Consequently, it is reasonable to suggest that other lncRNAs, such as HOTAIR, may also play a regulatory role in tumor metabolism.

Our initial analysis, utilizing data from The Cancer Genome Atlas (TCGA), revealed that HOTAIR is overexpressed in samples from patients with colorectal cancer and its expression increases with tumor stage. Furthermore, HOTAIR has been identified as a negative factor for overall survival. This finding aligns with the existing literature, which indicates that HOTAIR upregulation is associated with advanced tumor stages, metastasis, and poor prognosis [33]. Additionally, our examination of HOTAIR expression in cell lines confirmed this increase in comparison to non-tumor cells, consistent with findings reported by Pan et al. (2019) [34] and Huang et al. (2021) [35] in colorectal cancer cells.

On the other hand, in colorectal tumors, metabolic reprogramming represents an important process during carcinogenesis and it has been described that glycolysis is associated with the overexpression of key factors that include the PI3K/Akt/HIF-1α pathway [36]. However, despite the importance of the transcription factor HIF-1α, our data showed low expression in tumor tissues obtained from the TCGA, as well as a null effect on overall survival. Meanwhile, at the protein level, the data obtained from The Protein Atlas showed a high expression of HIF-1α, similar to that reported in multiple studies that used paraffin-embedded samples, and a 66.7% higher expression in tumor tissue compared to 12–25% in colorectal adenomas [37,38,39]. While in cell lines, we were able to demonstrate the overexpression of HIF1A compared to non-tumor cells and, in turn, its expression pattern was similar to that observed in HOTAIR, establishing a positive correlation, which could be associated with the constitutive activation of HIF-1α described by our group in cervical cancer cells, as well as by the induction of HOTAIR through the binding of *HIF1A* to its promoter [40,41]. 

Thus, the above served as a basis for evaluating the HOTAIR/HIF-1α axis, taking into consideration the role of the transcription factor as a master regulator of metabolism by inducing the expression of SLC2A1, HK1, and LDHA [24]. Furthermore, our protein–protein interaction analysis and the presence of a high number of hypoxia response elements in the promoters of these proteins suggested the possible regulatory effect of HIF-1α on most of these glycolytic enzymes and particularly on the glutamine transporter SLC1A5. While evaluating the endogenous mRNA expression levels of these enzymes, we observed that the RKO and SW-620 lines presented higher expression of HK2, PFKB4, PKLR, LDHA, PDHA, GLS2, GLUD1, and GOT1.

To investigate the regulatory effect of HOTAIR on specific enzymes, we employed a differential expression model. Based on this, we evaluated the percentage of cell viability. Our findings revealed significant changes associated with both the overexpression and inhibition of HOTAIR expression in RKO cells. Specifically, we observed that HOTAIR overexpression led to an increase in cell viability, a result that is consistent with findings reported in various cancer models [34,42]. Moreover, overexpression of HOTAIR leads in several cases to increased cell viability because this lncRNA regulates mechanisms that mediate the proliferation and evasion of apoptosis in tumour cells [43,44,45].

Regarding the increase in viability detection due to the inhibition of HOTAIR expression, it is important to highlight that the metabolic changes demonstrated in this article are likely to impact mitochondrial function and, consequently, the detection of MTT. Given that the metabolic processes regulated by HOTAIR tend to influence glutaminolysis in the RKO cell line, and considering that c-MYC is more highly expressed in this cell line (Appendix A), it is plausible to hypothesize that there are alterations in this pathway that affect glutathione production [46]. Studies have shown that a reduction in glutathione production results in lower levels of reactive oxygen species (ROS) [47]. These molecules have been found to directly affect formazan production, which is critical for MTT detection in viability assays [48]. This finding suggests that while there may not be a definitive method for assessing cell viability through HOTAIR inhibition, the MTT assay provides an alternative way of exploring the potential metabolic changes induced by HOTAIR, depending on the specific cell model used.

In addition, our results confirmed that the presence of HOTAIR influenced the mRNA expression of PFKB4, PGK1, LDHA, SLC1A5, GLUD1, and GOT1, which directly affected lactate and glutamate production. Although previous studies have identified HOTAIR as an inducer of SLC2A1 and HK2 expression in hepatocarcinoma, esophageal cancer, and glioblastoma [22,23,25,49,50], we did not find validation for these enzymes in more than two lines, except for the SW-620 line. Despite this discrepancy, our data demonstrated the role of HOTAIR in glucose metabolism through lactate production and the identification of other glycolytic enzymes. Furthermore, our findings are the first to demonstrate HOTAIR’s function in regulating glutaminolysis through the mRNA expression of the SLC1A5, GLUD1, and GOT1, having a direct impact on glutamate production.

Thus, increased lactate production may help meet bioenergetic demands, maintain redox homeostasis by reducing NADH production, induce angiogenesis, and facilitate epigenetic regulation through histone lactylation, which contributes to tumorigenesis [5,24,51]. Similarly, the rise in glutamate production can provide carbon and nitrogen for purine and pyrimidine synthesis and contribute to forming amino acids necessary for protein synthesis. Glutamate is also transformed into α-ketoglutarate, which acts as an intermediary in the Krebs cycle. Additionally, as previously mentioned, it supports the production of glutathione to combat oxidative stress and functions as a signaling molecule by activating the STAT3 transcription factor, promoting cell proliferation [51,52,53].

It is important to note that the effect of HOTAIR on the regulation of metabolic enzyme expression does not seem to be significant in HCT-116 cells. On the contrary, a substantial role has been observed in the SW-620 cell line, where both the overexpression and silencing of HOTAIR have a pronounced impact on the expression of several glycolytic enzymes, including SLC2A1 and HK2. This finding aligns with reports from other tumor types, highlighting its influence on glucose consumption and lactate production. This discrepancy could be related to the specific molecular subtypes of cancer present in these cell lines. Specifically, HCT-116 cells belong to the CMS1 subtype, characterized by microsatellite instability, a mutation in PI3K, and wild-type p53. On the other hand, the SW-620 line corresponds to the CMS2 subtype, which exhibits activation of the Wnt and c-MYC pathways, along with mutated p53 [54]. In addition to the above, it has been described that cells with metastatic potential mainly present an oxidative metabolism [53], so that aerobic glycolysis is highly active and corresponds to the type of metabolism presented by the SW-620 line since it comes from a metastasis, contrary to the primary tumor from which the HCT-116 and RKO lines originated.

While in the case of the RKO line, despite presenting a similar pattern to the SW-620 line in the expression of glycolytic and glutaminolic enzymes, as well as in the production of lactate and glutamine, there is low expression of HOTAIR but the production of lactate and glutamate is increased compared to SW-620 (Appendix A); this initially led us to hypothesize that HOTAIR did not play a role in the regulation of glutaminolysis. However, our differential expression model demonstrated that HOTAIR regulates the transcription of enzymes, such as SLC1A5, GLUD1, and GOT1, as well as glutamate production. Despite presenting similar characteristics to HCT-116 cells, except for the KRAS mutation, and in contrast to the presence of a BRAF mutation [55], HOTAIR seems to have greater importance in the RKO line, which could be explained through the constitutive activation of BRAF and the fact that it can induce the expression of HIF-1α and HIF-2α; the latter, in turn, can interact with c-MYC and, in this way, regulate glycolytic and glutaminolytic metabolism [55,56].

On the other hand, our initial hypothesis suggested that the effect of HOTAIR on the induction of metabolic enzyme expression could be mediated by the transcription factor HIF-1α. However, the ChiP assay showed that the regulation of *LDHA* and *GLUD1* expression was associated with the positioning of HIF-1α in the promoter of these enzymes only in the RKO line, despite the fact that there was no effect on the expression of the transcription factor mediated by HOTAIR, but it does confirm that the direct interaction of HOTAIR/HIF-1α observed in this line promotes the synthesis of these enzymes and is dependent on the increase in the expression of HOTAIR. Contrary to what was observed in the SW-620 line, where it was expected that the mechanism by which HOTAIR regulates the expression of metabolic enzymes was associated with HIF-1α due to its regulatory effect on the transcription factor itself, however, the positioning of HIF-1α in the *LDHA* promoter occurred when HOTAIR was overexpressed, similar to what showed in the HCT-116 line for the *SLC2A1* promoter. These results suggest that HOTAIR helps the positioning of HIF-1α in the promoters of some of its targets, similar to those described for other transcription factors, such as c-MYC [57]. Nevertheless, this effect was not observed under endogenous conditions, suggesting that HOTAIR could canonically regulate the expression of the enzymes identified in this work as *SLC1A5* through other transcription factors. As suggested by our bioinformatics analysis, which revealed the potential of c-MYC as a regulator of glycolytic and glutaminolytic enzymes [5,52], as well as the possible interaction between this transcription factor and HOTAIR, these findings provide an opportunity to explore the role of HOTAIR as a master regulator, operating through mechanisms that are independent of HIF-1α.

Additionally, the subcellular distribution of HOTAIR has been reported mainly in the nucleus, which is associated with the major function described for this lncRNA on transcriptional regulation in conjunction with the polycom repressor complex [58,59]. However, there are also findings indicating that HOTAIR can be detected in both the cytoplasm and the nucleus, suggesting a dual localization [5,60]. In this context, HOTAIR has been described as a “molecular sponge” regulating the stability and localization of miRNAs, as well as the translation of other RNAs [61]. Thus, the cellular compartmentalization observed in our models could exert another type of regulation mediated by HOTAIR, such as its activity as a miR-326/FUT6 sponge, which positively regulates Akt/mTOR signaling and may be dependent on the molecular subtype or tumor stage [34], which requires in-depth evaluation.

Furthermore, tumor cells can evade the inhibition of metabolic pathways by activating certain enzyme isoforms or upregulating alternative pathways that meet their energy or biomass needs [4]. This effect was observed by the use of the pharmacological combination of Doxorubicin/Metformin/Sodium Oxamate (3Tx), which has as its main targets the inhibition of Akt/HIF-1α signaling, complex I of the respiratory chain, and DNA synthesis [27,28,29]. It mainly generates the inhibition of glycolysis, which we demonstrate through the decrease in viability, the expression of HOTAIR, its glycolytic targets, glucose consumption, and ATP production. However, as a compensatory mechanism to the inhibition of this metabolic pathway, the tumor cell promoted the increase in glutamine consumption and glutamate production for the maintenance of lactate production through the Krebs cycle [62,63]. Despite this “resistance” effect, silencing HOTAIR reversed this overregulation of glutaminolysis, affecting proliferation, without ruling out the decrease in other biological processes, such as metastasis, considering that the SW-620 line presents a high migration potential and HOTAIR has been associated with metastatic potential [64].

Finally, the identification of a molecule that can regulate two of the most important and complementary metabolic pathways in tumor cells suggests that HOTAIR plays a crucial role in metabolic reprogramming. This discovery positions HOTAIR as a significant therapeutic target for cancer treatment by inhibiting abnormal metabolism.

## 5. Conclusions

In the present study, we demonstrate that HOTAIR participates in metabolic reprogramming by regulating the expression of several enzymes, including PFKFB4, PGK1, LDHA, SLC1A5, GLUD1, and GOT1, promoting the increase in lactate and glutamate production. Through the increase in the positioning of HIF-1α in the promoters of *LDHA* and *GLUD1*, without ruling out a similar regulatory mechanism associated with other transcription factors that regulate other HOTAIR-dependent enzymes. This allowed us to identify a potential target for the development of new therapeutic strategies focused on metabolic reprogramming, considering that HOTAIR is a molecule capable of regulating anaerobic glycolysis as well as glutaminolysis (Figure 7).

## Figures and Tables

**Figure 1 cells-14-00388-f001:**
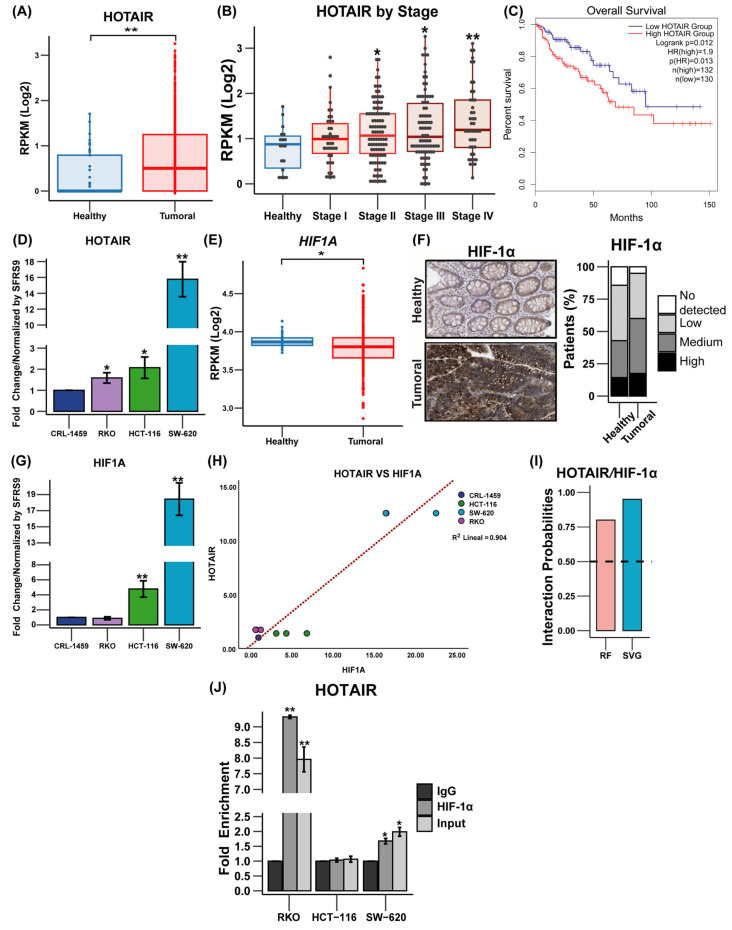
Expression of HOTAIR and HIF-1α in colorectal cancer tissues and cells. (**A**) Expression levels of HOTAIR in 41 non-tumor tissue samples vs. 471 tumor tissue samples (TCGA). (**B**) Relative expression of HOTAIR in healthy colorectal tissues and different degrees of carcinogenesis. (**C**) Kaplan–Meier survival curve of 132 colorectal cancer patients overexpressing HOTAIR vs. 130 patients with low HOTAIR expression (GEPIA 2). (**D**) mRNA expression of HOTAIR in colorectal cancer cell lines. (**E**) Expression levels of *HIF1A* in 41 non-tumor tissue samples vs. 471 tumor tissue samples (TCGA). (**F**) Immunohistochemistry of HIF-1α in non-tumor vs. tumor tissue. (**G**) mRNA expression of HIF1A in colorectal cancer cell lines. (**H**) Correlation of HOTAIR/HIF1A mRNA expression in colorectal cancer cell lines. (**I**) RPIseq analysis of HOTAIR interaction with HIF-1α. (**J**) RIP assay of HOTAIR/HIF-1α interaction. *HIF1A* corresponds to the gene and HIF-1α corresponds to the protein. Data are presented as mean ± SD. * *p* < 0.05, ** *p* < 0.005.

**Figure 2 cells-14-00388-f002:**
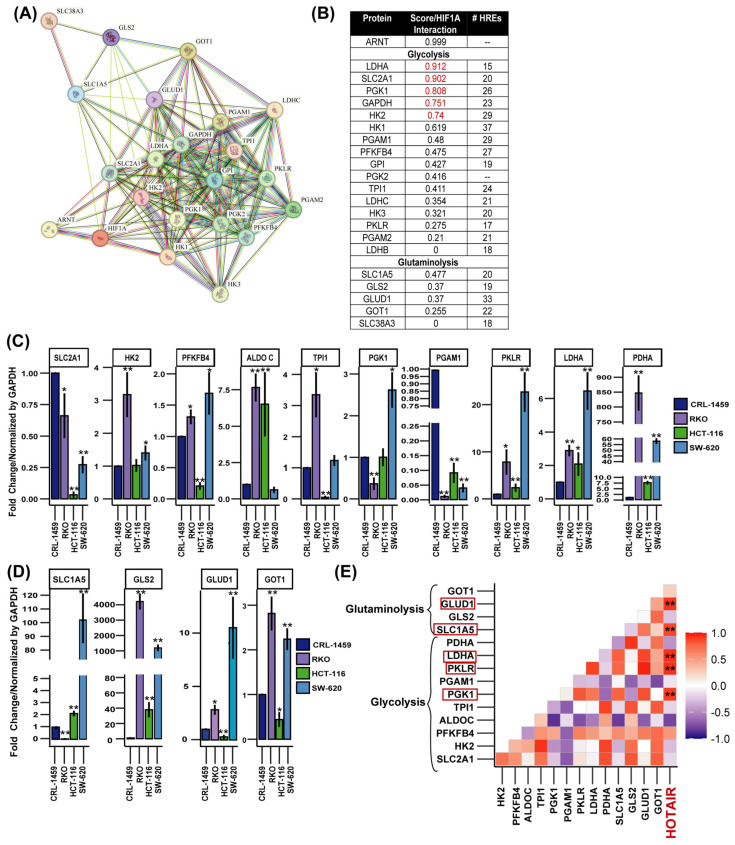
Expression levels of glycolytic and glutaminolytic enzymes in colorectal cancer cells and their possible association with the HIF-1α/HOTAIR axis. (**A**) Protein–protein interaction network of HIF-1α with glucose and glutamine metabolism enzymes. (**B**) Protein–Protein interaction score and number of HREs of HIF-1α with glycolytic and glutaminolytic enzymes. (**C**) mRNA expression levels of glycolytic enzymes in colorectal cancer cells. (**D**) mRNA expression levels of glutaminolytic enzymes in colorectal cancer cells. (**E**) Correlation of HOTAIR/glycolytic and glutaminolytic enzyme expression in colorectal cancer cell lines. HOTAIR is highlighted in red because enzyme expression levels were compared relative to its expression level. Data are presented as mean ± SD. * *p* < 0.05, ** *p* < 0.005.

**Figure 3 cells-14-00388-f003:**
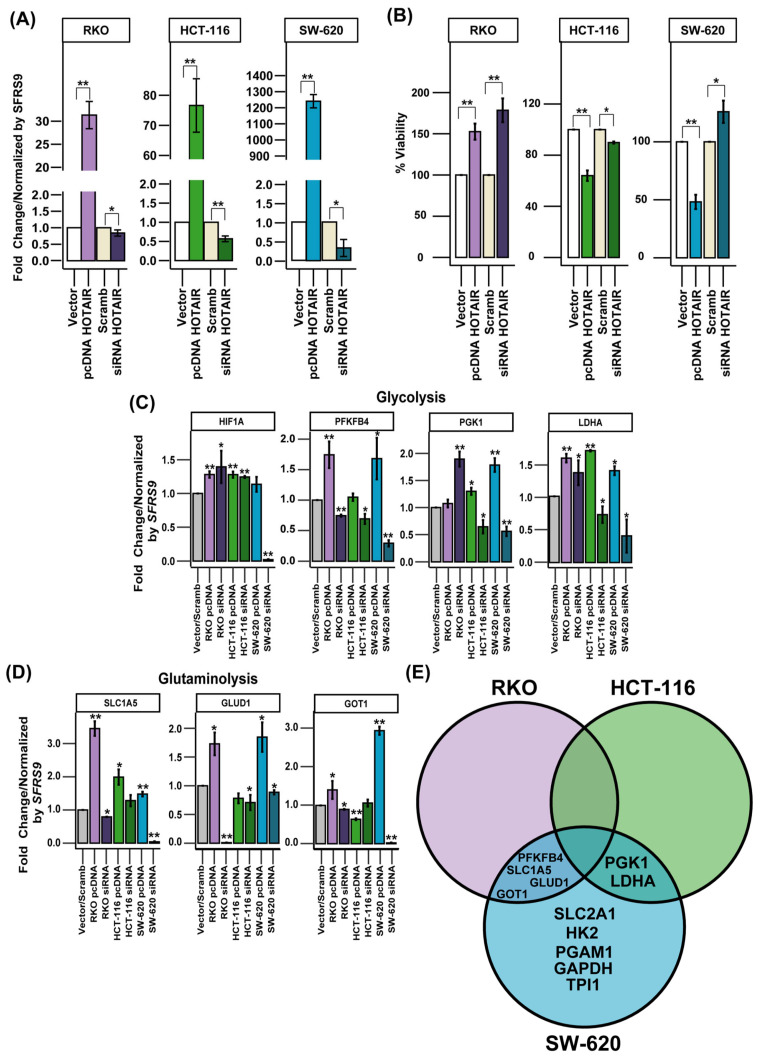
Expression levels of glycolytic and glutaminolytic enzymes regulated by HOTAIR. (**A**) Differential expression model of HOTAIR in colorectal cancer cells. (**B**) Viability of colorectal cancer cells after HOTAIR overexpression or silencing. (**C**) mRNA expression levels of glycolytic enzymes in colorectal cancer cells after HOTAIR overexpression or silencing. (**D**) mRNA expression levels of glutaminolytic enzymes in colorectal cancer cells after HOTAIR overexpression or silencing. (**E**) Venn diagram of differentially expressed enzymes after modification of HOTAIR expression in colorectal cancer lines. Data are presented as mean ± SD. * *p* < 0.05, ** *p* < 0.005.

**Figure 4 cells-14-00388-f004:**
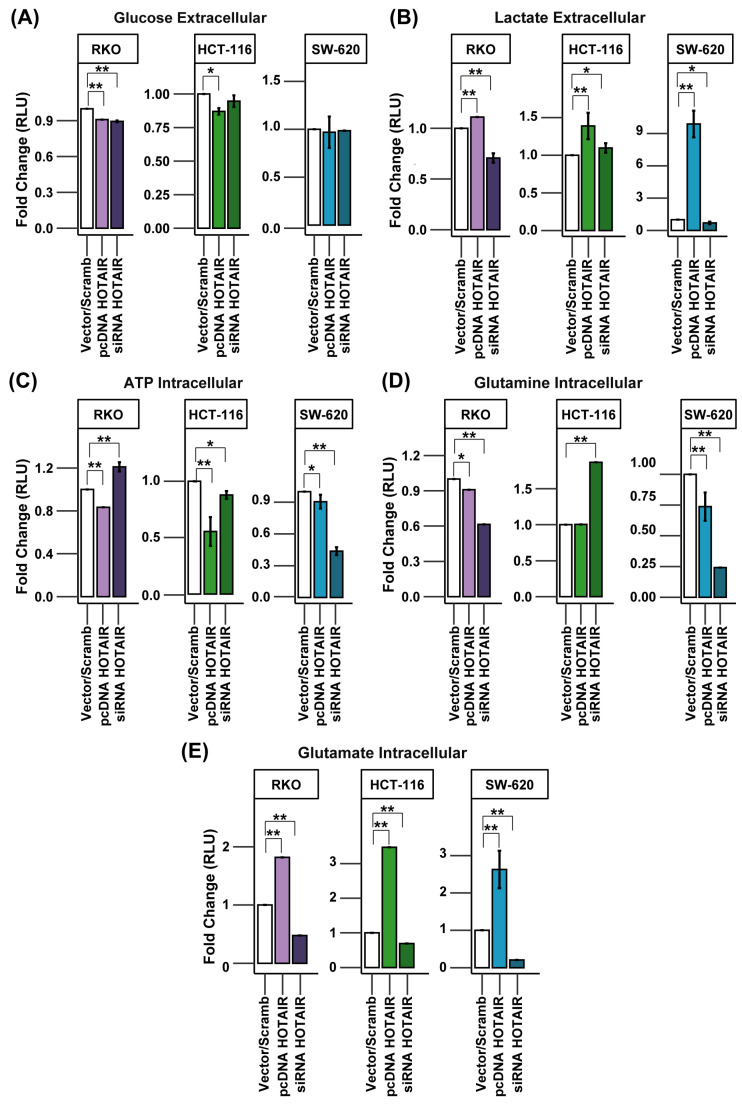
HOTAIR regulates lactate and glutamate production in colorectal cancer cells. (**A**) Glucose uptake in colorectal cancer cells after HOTAIR overexpression or silencing. (**B**) Lactate production in colorectal cancer cells after HOTAIR overexpression or silencing. (**C**) ATP production in colorectal cancer cells after HOTAIR overexpression or silencing. (**D**) Glutamine uptake in colorectal cancer cells after HOTAIR overexpression or silencing. (**E**) Glutamate production in colorectal cancer cells after HOTAIR overexpression or silencing. Data are presented as means ± SD. * *p* < 0.05, ** *p* < 0.005.

**Figure 5 cells-14-00388-f005:**
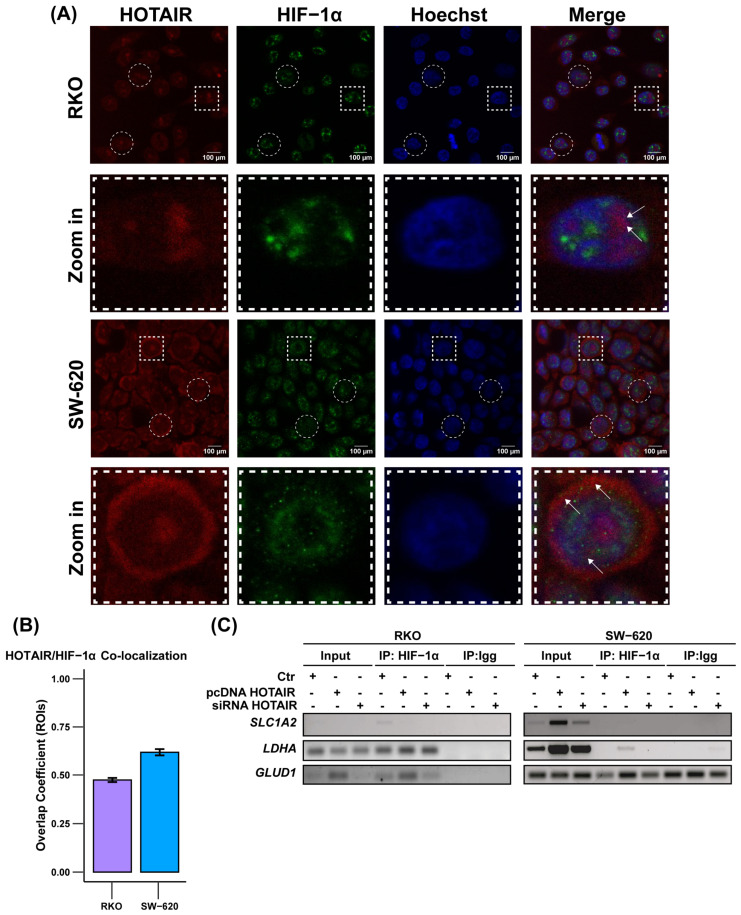
Cellular distribution of HOTAIR/HIF-1α. (**A**) Co-localization of HOTAIR and HIF-1α in colorectal cancer cells. Regions of Interest (ROIs), represented by circles and squares, were selected. The areas within the squares were then cropped and magnified for closer examination. Arrows indicated HOTAIR/HIF-1α co-localization (**B**) Overlap coefficient generated by n = 10 ROIs. Bar = 100 and 10 µm. (**C**) Positioning of HIF-1α at *LDHA* and *GLUD1* promoters by ChiP assay. Data are presented as means ± SD.

**Figure 6 cells-14-00388-f006:**
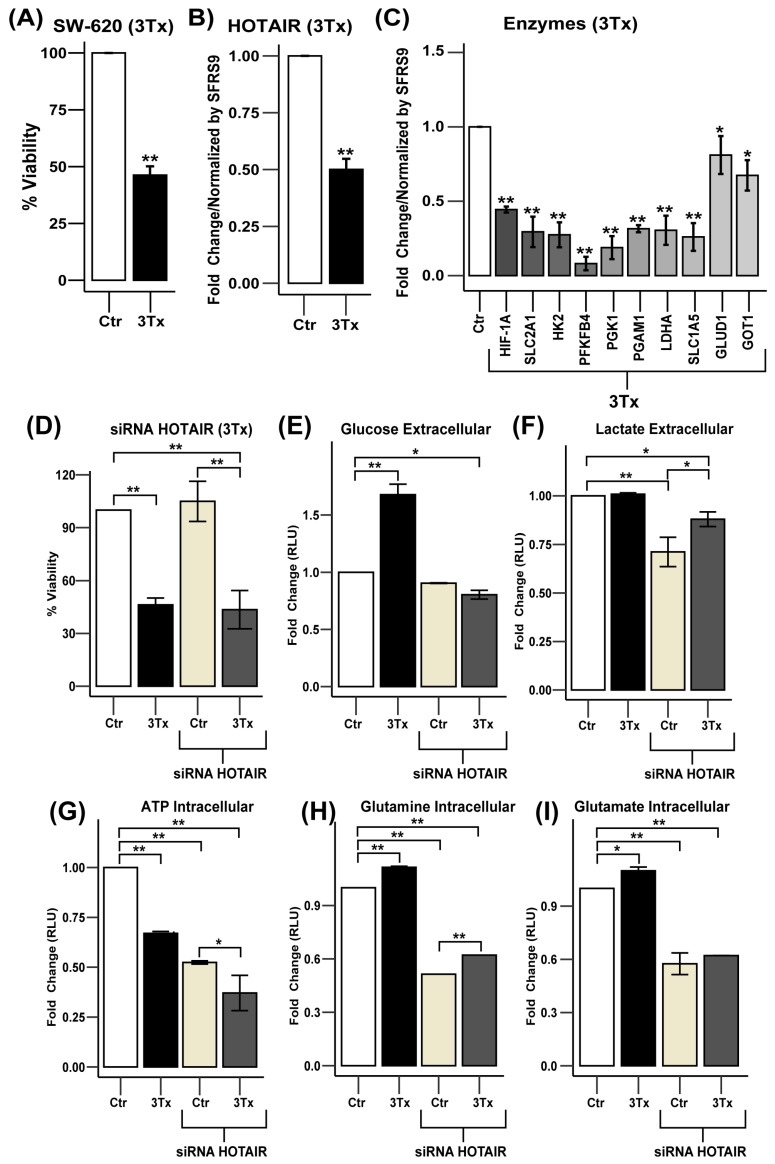
HOTAIR inhibition decreases glycolytic and glutaminolic enzyme expression in SW-620 cells. (**A**) Viability of SW-620 cells treated with triple therapy (3Tx). (**B**) HOTAIR expression levels in SW-620 cells treated with 3Tx. (**C**) mRNA expression levels of metabolic enzymes in SW-620 cells treated with 3Tx. (**D**) Viability of SW-620 cells treated with 3Tx with or without HOTAIR silencing. (**E**) Glucose uptake in SW-620 cells treated with 3Tx with or without HOTAIR silencing. (**F**) Lactate production in SW-620 cells treated with 3Tx with or without HOTAIR silencing. (**G**) ATP production in SW-620 cells treated with 3Tx with or without HOTAIR silencing. (**H**) Glutamine uptake in SW-620 cells treated with 3Tx with or without HOTAIR silencing. (**I**) Glutamate production in SW-620 cells treated with 3Tx with or without HOTAIR silencing. Data are presented as means ± SD. * *p* < 0.05, ** *p* < 0.005.

**Figure 7 cells-14-00388-f007:**
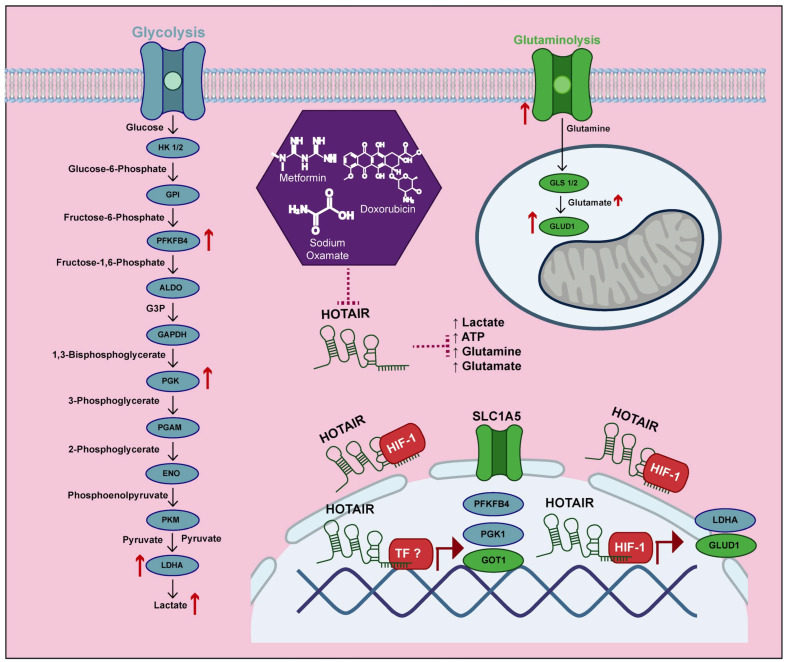
HOTAIR increases the production of lactate and glutamate by increasing the expression of glycolytic and glutaminolic enzymes.

**Table 1 cells-14-00388-t001:** qRT-PCR primer sequences.

Name	Sequence Forward	Sequence Reverse
HOTAIR	GGTAGAAAAAGCAACCACGAAGC	ACATAAACCTCTGTCTGTGAGTGCC
HIF1A	GAT AGT GAT ATG GTC AAT GAA TTC	GTC TGC TGG AAT ACT GTA ACT GTG
SFRS9	CCC TGC GTA AAC TGG ATG	ACC GAG ACC GTG AGT AGC C
SLC2A1	TGGCATCAACGCTGTCTTCT	CTAGCGCGATGGTCATGAGT
HK2	TCGCCGGTAGCCTTCTTTGT	AGAGATACTGGTCAACCTTCTGC
GPI	CGACTAGTGCACAGGGAGTG	CCATGGCGGGACTCTTGC
PFKFB4	GTCTCCAGCATCCTGCAAGT	TATCGATCTGGCCGTTCCTG
ALDOC	ATCGAGCAGTAACCAGTGGG	GCCACAAGAAGGACCTGAAG
TPI1	AGCAGACAAAGGTCATCGCA	CCAGTCACAGAGCCTCCATA
GAPDH	TCA AGA AGG TGG TGA AGC AG	AAA GGT GGA GGA GTG GGT GT
PKG1	TGGAGCTCCTGGAAGGTAAAG	GTTCCTGGCACTGCATCTCT
PGAM1	CCGGAATCTGCTAATCCCAGT	ACTCATAGCCAGCATCTCGTAG
PKLR	AGCATGTCGATCCAGGAGAAC	CCAGTAGGCAGAGGTGTTCC
LDHA	CCGGATCTCATTGCCACGC	CAAGTTCATCTGCCAAGTCCTTCA
SLC1A5	TTT TTC CTG GTC ACC ACG CTG	TCA TAG GTG GTA GAG TAT GAG CGA
GLS2	CAA GCT GGG GAA CAG CCA TA	GCT GAC AAG GCA AAC CTT CG
GLUD1	CCT GCA AGG GAG GTA TCC GT	ATG TCT GGA GCA GGC ACA TC
GOT1	TCG TGC GGA TTA CTT GGT CC	CTC AAC CTG CTT GGG GTT CA
SLC2A1-ChiP-1	GGCTCCACCATTTTGCTAGAGA	CGGACCGTAGCGTTTATAGGA
SLC2A1-ChiP-2	GCAAAAGCAAGGCTTGGCTC	TGGGTGACTTCGGTGCACTA
LDHA-ChiP-1	GCACCTTACTTAGACTCCCAGCG	CGGGAGGGGCCTTAAGTGGA
LDHA-ChiP-2	AGGCTTCACTGTGAGTGGGAGC	GGGAGGTTACTCTCAGGAAGGC
SLC1A5-ChiP-1	GGAACGAACCCCTGTGGTTTAAGG	TGCAGAGCGTTCGGAGACTGGA
SLC1A5-ChiP-2	GGGTTTCACCATGTTGGCCAGG	GCCCTGAGTTTGGTCTTTAGTCG
GLUD1-ChiP-1	GCACATACCTGAGAGCCCCG	AGGACGGACTTCGGGGACAG
GLUD1-ChiP-2	GCTTTCCTGCCCACGTGTCAGC	TGTAGGGCGCTCAGAGGCCGA

## Data Availability

Any data will be distributed upon request to carlos.pplas@gmail.com and lcflores@hotmail.com.

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
