# Peer review of "HOTAIR Participation in Glycolysis and Glutaminolysis Through Lactate and Glutamate Production in Colorectal Cancer"

_cells, 2025, doi:10.3390/cells14050388_

Round 1
Reviewer 1 Report
Comments and Suggestions for Authors
Summary
This manuscript provides important insights into how the long non-coding RNA HOTAIR contributes to metabolic reprogramming in colorectal cancer cells. By demonstrating that HOTAIR regulates multiple enzymes in glycolysis and glutaminolysis (e.g., PFKFB4, PGK1, LDHA, SLC1A5, GLUD1, GOT1), the authors underscore a potential master regulatory role for HOTAIR in tumor bioenergetics. The findings presented here could be valuable for understanding the mechanisms behind “oncometabolite” production and may inspire novel therapeutic strategies.
On the basis of the data and the discussion, I consider this work to be of high quality and recommend acceptance, provided the authors address the following major points and incorporate the suggested improvements.
Major Comments
- Cell-Line-Specific Mechanisms and Transcription Factor Involvement
While the authors clearly show that HOTAIR exerts differential metabolic regulation in RKO, HCT-116, and SW-620 cells, further discussion (and possibly limited additional data, if feasible) would help clarify why some enzymes are more strongly affected in one cell line than in another. This could be tied to known mutations (KRAS, BRAF, p53) or CMS subtypes. In addition, deeper consideration of other transcription factors (e.g., c-Myc), especially where HIF-1α may not fully account for the results, would strengthen the mechanistic interpretation.
- Functional Impact of Metabolite Changes and Therapeutic Implications
The authors demonstrate changes in lactate and glutamate production driven by HOTAIR, but they could more explicitly connect these findings to cell viability, proliferation, or drug response. For instance, it would be valuable to test whether HOTAIR knockdown synergizes with the 3Tx combination or affects cellular ROS levels. This would clarify whether metabolic reprogramming mediated by HOTAIR has a direct bearing on the therapeutic potential of targeting lncRNAs in colorectal cancer.
Minor Comments
Language and Gene Name Consistency
Throughout the manuscript, please ensure consistent use of gene/protein names (e.g., SLC2A1 vs. GLUT1, SLC1A5) and correct any minor typos (e.g., “Kaplan–Meier” instead of “Kapla-Meier,” “overexpressed” instead of “sobreexpresed”). A quick proofreading pass for spelling and grammar will help the clarity of presentation.
Overall Recommendation
In conclusion, I strongly recommend accepting this manuscript after the authors address the Major Comments outlined above. The work is timely, insightful, and should be of high interest to readers of Cells. Addressing these points will ensure the study’s contributions are clearly positioned and fully justified, thereby enhancing its impact and clarity.
Comments on the Quality of English LanguageCheck for inconsistent use of terms and gene name notation
For example, check that there is no mixture of gene and protein names, such as where GLUT1 is written as “SLC2A1” or where “SLC1A2” appears (in particular, where it says “SLC1A2 (GLUT1)” but in fact SLC2A1 is the official name for GLUT1).
As “GLUD1”, “GLS2”, “GOT1”/“GOT2” etc. also appear at the same time, it would be easier to understand if the gene names and protein names were clearly distinguished in the text, or if the same name was always used.
Details of words (English notation)
“Kapla-Meier” → “Kaplan–Meier”
“sobreexpresed” → “overexpressed”
“Assessment coefficient” → For example, “Correlation coefficient” would be appropriate.
As there are a number of minor spelling errors, we recommend that you check the spelling and grammar of your English before submitting.
Author Response
- While the authors clearly show that HOTAIR exerts differential metabolic regulation in RKO, HCT-116, and SW-620 cells, further discussion (and possibly limited additional data, if feasible) would help clarify why some enzymes are more strongly affected in one cell line than in another. This could be tied to known mutations (KRAS, BRAF, p53) or CMS subtypes. In addition, deeper consideration of other transcription factors (e.g., c-Myc), especially where HIF-1α may not fully account for the results, would strengthen the mechanistic interpretation.
Dear reviewer, thank you for your kind observation. The differential effects of HOTAIR in colorectal cancer cell lines, as discussed on line 535, may be related to the specific mutations present in each cell line, with BRAF mutations being particularly significant. Additionally, a key difference among the cell lines analyzed in our study is their origin, which affects their metastatic potential. This, in turn, has a direct impact on endogenous metabolism, a factor of particular interest for the SW-620 line. We discuss these concepts to clarify the results presented in the paper (lines 499-511).
On the other hand, we agree with the reviewer that further study of other transcription factors associated with metabolic reprogramming is needed. To this end, in this paper, we decided to perform a primary bioinformatics analysis addressing the role of c-MYC and its possible association with HOTAIR shown in Supplementary Figure 6 and, lines 390-400 and 580-584. This serves as a reference point for subsequent analysis of this transcription factor and its role in aberrant metabolism through possible interaction with HOTAIR
- Functional Impact of Metabolite Changes and Therapeutic Implications. The authors demonstrate changes in lactate and glutamate production driven by HOTAIR, but they could more explicitly connect these findings to cell viability, proliferation, or drug response. For instance, it would be valuable to test whether HOTAIR knockdown synergizes with the 3Tx combination or affects cellular ROS levels. This would clarify whether metabolic reprogramming mediated by HOTAIR has a direct bearing on the therapeutic potential of targeting lncRNAs in colorectal cancer.
After considering the recommendation of Reviewer 1, we completed Figure 6 to illustrate the therapeutic implications of HOTAIR as a modulator of tumor metabolism. We gathered additional information on the use of triple therapy (3Tx) and its synergistic effect when combined with HOTAIR silencing (lines 422-448). This information demonstrated how tumor cells can evade the inhibition of a metabolic pathway and highlighted the synergistic effect observed after the silencing of HOTAIR (lines 596-609). These results underscore the significant potential of inhibiting lncRNA when used in conjunction with drugs designed to counteract abnormal metabolic processes
- Language and Gene Name Consistency
Throughout the manuscript, please ensure consistent use of gene/protein names (e.g., SLC2A1 vs. GLUT1, SLC1A5) and correct any minor typos (e.g., “Kaplan–Meier” instead of “Kapla-Meier,” “overexpressed” instead of “sobreexpresed”). A quick proofreading pass for spelling and grammar will help the clarity of presentation.
Dear reviewer, we appreciated your insightful comment. The nomenclature of the genes and proteins evaluated in this study was standardized in the text and all figures. Spelling was checked and typographical errors were corrected.
Reviewer 2 Report
Comments and Suggestions for Authors
Dear editor,
Authors aimed to evaluate the HOTAIR, a LncRNA regulates the expression of different genes. The paper is interesting and of interest, however, there are several points to be revised before being considered for publication.
- The last paragraph of the introduction section is summarizing the findings of the study. In this form, the paragraph is like and abstract. Author should revise this paragraph as merely indicating the aim of the study and rationale behind it.
- Line 89 CO2 should be CO2 with subscripted 2.
- There is no table legend for Table 1.
- Alexa 488 is not the name of the fluorophore. It is Alexa Fluor® 488 (line 162).
- Gene and protein nomenclature should be revised throughout the manuscript. Genes should be italicized. It is very hard to follow in this way.
- Manuscript needs to be revised in terms of language (i.e. use of capital letters).
- Why do both knockdown and overexpression of HOTAIR lead to elevated cell viability of RKO? Is it because of the use of MTT assay instead of using proliferation assay such as trypan blue dye exclusion assay?
Author Response
- The last paragraph of the introduction section is summarizing the findings of the study. In this form, the paragraph is like and abstract. Author should revise this paragraph as merely indicating the aim of the study and rationale behind it.
Dear reviewer, thank you for your positive comment. We revised the final paragraph of the introduction. This revision clarifies the study's objective, highlights the most significant findings, and explains their impact and justification (lines 72-80)
- Line 89 CO2 should be CO2 with subscripted 2.
Corrected as requested
- There is no table legend for Table 1
Dear reviewer the legends and citations in the text of each of the tables and figures contained in the manuscript were reviewed and added (line 118).
- Alexa 488 is not the name of the fluorophore. It is Alexa Fluor® 488 (line 162).
The trade name of the Alexa Fluor® 488 fluorophore and of each of the reagents used in the study was corrected (line 162).
- Gene and protein nomenclature should be revised throughout the manuscript. Genes should be italicized. It is very hard to follow in this way.
The nomenclature for the genes and proteins examined in this study was standardized throughout the text and all figures. Genes are italicized when referring to DNA, while proteins are presented in a regular font. Additionally, the mRNA nomenclature associated with the evaluated proteins is also formatted in a standard manner. The text includes references to the specific type of molecule corresponding to each name.
- Manuscript needs to be revised in terms of language (i.e. use of capital letters).
Spelling was checked and typographical errors were corrected.
- Why do both knockdown and overexpression of HOTAIR lead to elevated cell viability of RKO? Is it because of the use of MTT assay instead of using proliferation assay such as trypan blue dye exclusion assay?
The antioxidant system generated by glutathione is directly influenced by glutaminolysis. Therefore, it is reasonable to hypothesize that the inhibition of HOTAIR in the RKO cell line has a direct effect on glutamate production. This change may subsequently impact glutathione production, leading to a diminished response to reactive oxygen species (ROS). Previous studies have indicated that this diminished response affects formazan production, which can interfere with MTT detection; hence, it should not be considered a reliable indicator of increased cell viability. These concepts are discussed to provide clarity on the results presented in the article (lines 497-515).
After addressing the reviewers' comments, we believe that this revised manuscript is suitable for publication in the special issue “Non-Coding and Coding RNAs in Targeted Cancer Therapy” of Cells. Our work contributes to the understanding of the role of lncRNAs in the metabolic reprogramming of tumor cells and highlights the significant potential of these molecules as therapeutic targets.
Round 2
Reviewer 2 Report
Comments and Suggestions for Authors
Authors responded my comments appropriately. I believe that the manuscript can be accepted for publication.